# Identification and Validation of Reference Genes for Expression Analysis Using qRT-PCR in *Cimex hemipterus* (Hemiptera: Cimicidae)

**DOI:** 10.3390/insects13090784

**Published:** 2022-08-30

**Authors:** Delong Kong, Daxia Shi, Changlu Wang, Ruyue Zhai, Lingling Lyu, Yurong He, Desen Wang

**Affiliations:** 1Department of Entomology, South China Agricultural University, Guangzhou 510642, China; 2Key Laboratory of Bio-Pesticide Innovation and Application of Guangdong Province, Guangzhou 510642, China; 3Engineering Research Center of Biological Control, Ministry of Education, Guangzhou 510642, China; 4Department of Entomology, Rutgers University, New Brunswick, NJ 08901, USA; 5Ningbo Dayang Technology Co., Ltd., Ningbo 315000, China

**Keywords:** *Cimex hemipterus*, reference gene, quantitative real-time PCR

## Abstract

**Simple Summary:**

Reference genes are the basis for exploring the function of target genes in insects. Recently, we identified a large number of chemoreceptor genes by transcriptome sequencing in different tissues of *Cimex hemipterus* (F.) (Hemiptera: Cimicidae). Additionally, we plan to study the expression patterns of these genes and clarify their functions. However, there is no report on the screening and application of the reference genes in *C*. *hemipterus*. In the present study, the expression-level stability of 10 candidate reference genes in *C. hemipterus* under various experimental conditions was evaluated through qRT-PCR combined with RefFinder (which integrates four computational programs: geNorm, NormFinder, BestKeeper, and ∆Ct). Results show that the optimal combination of reference genes for each experimental condition was as follows: *RPL8* and *EF1α* for the developmental stage, *RPL8* and *RPS16* for adult sex, *RPL8* and *RPL11* for adult tissue, *RPL8* and *β-tubulin* for gas stimulation, and *RPL8* and *NADH* for temperature. The results will lay a foundation for the follow-up study on the expression and function of the target genes of tropical bed bugs.

**Abstract:**

The tropical bed bug, *Cimex hemipterus* (F.) (Hemiptera: Cimicidae) is an important public-health pest that feeds on the blood of humans and some other animals. To explore the function of the target genes of *C. hemipterus*, it is essential to select suitable reference genes for the accurate quantification of gene expression. Here, we selected 10 frequently used reference genes in insects and evaluated their stability in *C. hemipterus* under various biotic (developmental stage, sex, and tissue) and abiotic (gas stimulation and temperature) conditions through RefFinder (which integrates four computational programs: geNorm, NormFinder, BestKeeper, and ∆Ct). Results indicate that the optimal combination of reference genes for each experimental condition was as follows: *RPL8* and *EF1α* for the developmental stage (eggs, early instar nymphs, late instar nymphs, and adults), *RPL8* and *RPS16* for adult sex, *RPL8* and *RPL11* for adult tissue (head, thorax, abdomen, and legs), *RPL8* and *β-tubulin* for gas stimulation (air and carbon dioxide), and *RPL8* and *NADH* for temperature (0, 5, 17, 30, and 37 °C). Finally, the expression pattern of the *HSP70* and *GR21* genes were analyzed, and the results highlight the importance of appropriate reference-gene selection. Our results provide a comprehensive list of optimal reference genes from *C. hemipterus* for the first time, which will contribute to accurately analyzing the expression of target genes.

## 1. Introduction

Quantitative real-time reverse transcription-polymerase chain reaction (qRT-PCR) testing has been widely used in the detection and quantitative analysis of mRNA transcription levels, owing to its advantages of high accuracy, high specificity, high sensitivity, easy performance, and rapidity [1,2]. Nevertheless, the precision of qRT-PCR analysis is often influenced by many biological and technical factors, for example, RNA integrity, quality, and reverse transcription efficiency, as well as PCR efficiency [2,3]. To avoid errors caused by the above variables and obtain accurate and reliable results, reference genes are often applied to normalize the expression of target genes [4].

Generally speaking, the suitable reference gene should not be controlled or affected by the experimental procedure [5]. However, many studies indicated that any reference gene is also not guaranteed to be stably expressed under all experimental conditions [6,7,8,9,10,11,12,13]. Therefore, it is necessary to select suitable reference genes under specific conditions for certain insect species rather than using common reference genes [14].

Bed bugs, *Cimex hemipterus* (F.) and *Cimex lectularius* L. (Hemiptera: Cimicidae), are cryptic and nocturnal obligate blood-sucking insects that feed on the blood of humans and some other warm-blooded animals [15,16]. In China, both *C. hemipterus* and *C. lectularius* are distributed, among which southern provinces such as Guangdong, Guangxi, Fujian, and Hainan are dominated by *C. hemipterus*, while other provinces are dominated by *C. lectularius* [17,18,19,20]. Since 2007, the number of reports of bed bug infestations has increased dramatically in China [21,22,23].

Similar to other blood-sucking insects [24,25,26], bed bugs can accurately locate their hosts and successfully feed on blood by sensing host-derived signaling substances such as CO_2_, host odor, and heat [27,28,29]. At present, the intrinsic mechanism of mosquito perception of host-derived signaling substances has been extensively studied, and numerous odorant receptors (ORs), gustatory receptors (GRs), and ionotropic receptors (IRs) involved in chemosensory signal transduction have been identified [24,25,30,31,32]. For bed bugs, there are relatively few studies on the chemosensory mechanisms of *C. lectularius*, and these mainly focus on the olfactory sensing mechanisms of aggregated pheromones and host odors, as well as repellent substances [29]; however, studies on the chemosensory mechanism of *C. hemipterus* have not been reported yet. Exploring the internal mechanism of bed bugs’ perceptions of host-derived signal substances and identifying the relevant chemoreceptor genes will help enrich our comprehensive understanding of the bugs’ host-search behavior and provide an important theoretical basis for the timely monitoring of bed bugs. On the other hand, it will lay the foundation for seeking new bed bug control strategies, which is of great practical significance for the realization of safe and efficient bed bug control.

Recently, we identified a large number of chemoreceptor genes by transcriptome sequencing in different tissues of *C. hemipterus*. Additionally, we plan to study the expression patterns of these genes (for example, in different tissues and in different developmental stages of bugs) and clarify their functions. However, there is no report on the screening and application of the reference genes in *C. hemipterus*. Although reference genes have been reported in *C. lectularius* [33,34], which belongs to the same genus as *C. hemipterus*, it is unknown whether these reference genes are equally suitable for *C. hemipterus*. Thus, the objective of the present study was to screen the stably expressed reference genes of *C. hemipterus* under five experimental conditions (developmental stage, sex, tissue, gas (air and carbon dioxide) stimulation, and temperature) for qRT-PCR normalization. Our results will provide the basic information for subsequent target-gene-expression analysis and functional studies in *C. hemipterus* and then lay the foundation for a better understanding of this pest and the formulation of rational control methods.

## 2. Materials and Methods

### 2.1. Insects

The *Cimex hemipterus* used in this study were collected from Guangzhou City, Guangdong Province, China in 2018 [19]. They were maintained in artificial climate chambers (26 ± 1 °C, 40 ± 10% relative humidity (RH) with a photoperiod of 12:12 (L:D) h) according to the methods of Zhang et al. (2021) [35].

### 2.2. Experimental Condition and Sample Collection

A total of five experimental conditions (developmental stage, sex, tissue, gas stimulation, and temperature) were set up to assess the stability of the 10 candidate reference genes in *C. hemipterus*. For each replication, the developmental-stage sample includes 50 eggs, 20 1st–2nd instar nymphs, 8 4th–5th instar nymphs, and 5 male adults; the sex sample contains 5 male adults and 5 female adults; the body tissue sample includes the head (containing antennae and mouthparts), thorax, abdomen, and legs, which were dissected from 50 male adults; the gas stimulation sample contains 5 CO_2_-treated (the concentration of CO_2_ was 1.6%) male adults and 5 air-treated male adults—all the male adults were stimulated by CO_2_ or air for 3 h with a flow rate of 500 mL/min, respectively; the temperature treatment sample contains male adults that were exposed to 0, 5, 17, 30, and 37 °C for 2 h, with 4 individuals per temperature treatment. Each experimental condition was replicated three times. The collected samples were immediately frozen with liquid nitrogen and temporarily stored under the condition of −80 °C until RNA extraction.

### 2.3. RNA Extraction and cDNA Synthesis

TRIzol reagent (Takara Biomedical Technology (Beijing) Co., Ltd., Beijing, China) was used for extracting total RNA from each sample following the manufacturer’s manual. The concentration and purity of RNAs were determined by a NanoDrop One Spectrophotometer (Thermo Fisher Scientific, Waltham, MA, USA). The OD260/280 value of the total RNA was between 1.9 and 2.1, which was considered to meet the quality requirements for subsequent experiments. Then, the integrity of RNAs was examined by 0.8% agarose gel electrophoresis. First-strand cDNAs were synthesized by reverse transcribing 1 µg of total RNA with the PrimeScript RT Kit (Takara Biomedical Technology (Beijing) Co., Ltd., Beijing, China).

### 2.4. Reference Gene Selection, Primer Design, and qRT-PCR

Ten candidate reference genes were selected based on our recently sequenced transcriptomes for *C. hemipterus* (Kong et al., unpublished data). The 10 reference genes were *Ribosomal protein L8* (*RPL8*), *Ribosomal protein L11* (*RPL11*), *Ribosomal protein L13* (*RPL13*), *Ribosomal protein S16* (*RPS16*), *α-tubulin* (*α*-*tubulin*), *β-tubulin* (*β-tubulin*), *Glyceraldehyde 3-phosphate dehydrogenase* (*GAPDH*), *Elongation factor 1 α* (*EF1α*), *NADH dehydrogenase (ubiquinone) flavoprotein 1* (*NADH*), and *Actin* (*Actin*). Primer pairs for amplification were designed strictly according to the principles of qRT-PCR primer design using the software Primer Premier 5.0. The primer sequence, length, and amplification efficiency (E) of the 10 reference genes are listed in Table 1.

The 20 µL reaction system included 10 µL 2 × SYBR Green Premix (TIANGEN, China), 0.6 µL of 10 µM primers, 6.8 µL ddH_2_O, and 2 µL diluted cDNA. Reactions were conducted using a CFX-96 real-time PCR system (Bio-Rad, Hercules, CA, USA). The amplification program was as follows: 95 °C for 10 min, 40 cycles of 95 °C for 10 s, and 60 °C for 32 s. The presence of a single peak in the melting curve analysis was selected to confirm the gene-specific amplification, as well as to get rid of the non-specific amplification and the primer–dimer generation.

The amplification efficiency and correlation coefficient of each candidate reference gene were measured by applying the slope of the standard curve generated by the serial five-fold diluted cDNA template [1]. The amplification efficiency (E) of each primer pair was calculated according to the equation [5]:E = (10^[−1/slope]^ − 1) × 100

### 2.5. Determination of Reference Gene Expression Stability

The expression stability of the 10 candidate reference genes was assessed using RefFinder (https://www.heartcure.com.au/reffinder/, accessed on 10 July 2022), which integrates four major computational programs, i.e., geNorm [36], NormFinder [7], BestKeeper [4], and the ∆Ct method [37]. The suitable number of reference genes for normalizing target gene expression was determined by computing the pairwise variation (V) using geNorm [36]. If the V-value (Vn/Vn+1) is lower than 0.15, it indicates using n genes is enough for target gene expression normalization.

### 2.6. Validation of the Candidate Reference Genes

To further verify the accuracy of our experimental results, the expression profiles of the heat shock protein (*HSP70*) gene and gustatory receptor 21 (*GR21*) gene in *C. hemipterus* were normalized by the top two most stable (*RPL8* and *RPL11*) and least stable (*RPL13* and *RPS16*) candidates in the head (including antennae and mouthparts), thorax, abdomen, and legs. The forward primer of *HSP70* was 5′-GAACGGGAACCAAGGAGGC-3′, and the reverse primer of *HSP70* was 5′-TCGACGGCGAATCTCAGCA-3′. The forward primer of *GR21* was 5′-CGGGCATGGTGTTTATCAAG-3′, and the reverse primer of *GR21* was 5′-CTCCCAGTGTAACTCGAGGG-3′. The relative expression levels of *HSP70* and *GR21* in different tissues (considered as different treatments) were computed using the 2^−∆∆Ct^ method [38]. Each treatment was replicated 3–5 times.

One-way analysis of variance (ANOVA) was used to analyze the expression differences between *HSP70* and *GR21* in different tissues, and Tukey’s honest significant difference (HSD) test was selected for separating the means. Statistical analysis was conducted using SPSS version 22.0 [39].

## 3. Results

### 3.1. Primer Specificity and Efficiency

Melting curves of all primer pairs were a single peak, and no nonspecific amplification products were observed, indicating that the primers were highly specific. The amplification efficiency (E) of the primers for the 10 reference genes ranged from 87.11% for *NADH* to 111.07% for *RPL8*, and the correlation coefficients varied from 0.9806 to 0.9990 (Table 1).

### 3.2. Ct Values of Candidate Reference Genes

The cycle threshold (Ct) values of the 10 reference genes under five experimental conditions were used for evaluating their expression levels. As shown in Figure 1 and Table 2, the Ct values ranged from 14.92 to 32.34, and the most values were distributed between 18 and 27. *Actin* was the most abundant transcript (the mean Ct value was 16.46 ± 0.40), and *NADH* was the least abundant candidate reference gene (the mean Ct value was 24.61 ± 0.56) (Table 2).

### 3.3. Stability of the 10 Reference Genes under Different Experimental Conditions

For the developmental stage, there was a significant difference between the two most stable reference genes recommended by the four algorithms (Table 3). In the geNorm analysis, *RPL8* and *RPL11* had the best stability. In the NormFinder analysis, *EF1α* and *NADH* were the two most stable genes. The BestKeeper analysis showed that *NADH* and *β-tubulin* were the top two ranked genes. Additionally, in the ∆Ct analysis, *RPL8* and *EF1α* were identified as the two most stable genes. Based on the RefFinder analysis results, *RPL8* was the most stable reference gene in the developmental stage condition, followed by *EF1α*, *NADH*, *RPL11*, *GAPDH*, *RPS16*, *β-tubulin*, *RPL13*, *α-tubulin*, and *Actin* (Figure 2). The pairwise variation value of V2/3 calculated by geNorm was below the threshold value of 0.15 (Figure 3). Thus, in different developmental stages, *RPL8* and *EF1α* were the suitable combination of reference genes.

For bed bug sex, the two most stable genes from the four computational programs differ significantly. For example, *RPS16* and *EF1α* were determined to be the top two most stably expressed referenced genes by geNorm, *RPL8* and *RPL13* for NormFinder, *RPS16* and *RPL11* for BestKeeper, and *RPL8* and *RPL11* for ΔCt (Table 3). Based on the integrated analysis results of RefFinder, the most stable reference gene in different sex conditions was *RPL8*, followed by *RPS16*, *RPL11*, *EF1α*, *GAPDH*, *RPL13*, *α-tubulin*, *Actin*, *β-tubulin*, and *NADH* (Figure 2). The results of the geNorm analysis indicate that the pairwise variation value of V2/3 was less than the threshold value of 0.15 (Figure 3). Therefore, the combination of *RPL8* and *RPS16* was recommended for normalizing the qRT-PCR data in the different sexes of adult samples by geNorm.

For bed bug tissue, geNorm and NormFinder had the same top two genes in stability ranking: *RPL8* and *RPL11* (Table 3). However, the top two in BestKeeper were *RPL8* and *α-tubulin*, and the top two in ΔCt were *EF1α* and *RPL8* (Table 3). The RefFinder analysis showed that *RPL8* was the most stable reference gene under this experimental condition, followed by *RPL11*, *EF1α*, *α*-*tubulin*, *GAPDH*, *Actin*, *β*-*tubulin*, *NADH*, *RPL13*, and *RPS16* (Figure 2). The pairwise variation value of V2/3 was less than the cut-off value of 0.15 according to the geNorm analysis. (Figure 3). Therefore, *RPL8* and *RPL11* were recognized as suitable reference genes for normalizing qRT-PCR data in the different tissue samples.

For gas stimulation, NormFinder and ΔCt had the same top two gene stability rankings: *RPL8* and *β*-*tubulin* (Table 3). However, based on geNorm, *Actin* and *β-tubulin* were the top two most stable genes. *EF1α* and *GAPDH* were determined to be the top two most stably expressed genes by BestKeeper (Table 3). Based on the analysis results of RefFinder, the most stable gene under the treatment of gas stimulation was *RPL8*, followed by *β*-*tubulin*, *Actin*, *RPL11*, *EF1α*, *α*-*tubulin*, *GAPDH*, *NADH*, *RPL13*, and *RPS16* (Figure 2). The geNorm analysis results indicated that the pair-wise value of V2/3 was lower than 0.15 (Figure 3), suggesting that the combination of *RPL8* and *β*-*tubulin* was sufficient for reliable normalization in different gas stimulation samples.

For temperature treatment, the top two most stable genes from the four computational programs differed from each other. They were *RPL8* and *RPL13* by geNorm, *RPL8*, and *β*-*tubulin* by NormFinder, *NADH* and *Actin* by BestKeeper, and *RPL8* and *NADH* by ΔCt (Table 3). Based on the RefFinder analysis, the most stable reference gene under the conditions of different temperatures was *RPL8*, followed by *NADH*, *β*-*tubulin*, *RPL13*, *EF1α*, *GAPDH*, *RPL11*, *Actin*, *α*-*tubulin*, and *RPS16* (Figure 2). The pairwise variation value of V2/3 calculated by the geNorm analysis program was less than 0.15 (Figure 3), indicating that the combination of *RPL8* and *NADH* was adequate for optimal normalization in different temperature treatment samples.

### 3.4. Validation of Candidate Reference Genes in Different Tissues

The expression trend of *HSP70* in different tissues was similar in the two combinations, with higher expression in the thorax and abdomen, and lower expression in the head and legs. When the two most stable reference genes (*RPL8* and *RPL11*) were selected as reference genes, the relative expression levels of *HSP70* in the legs were significantly higher than in the head (F_3,16_ = 17.1, *p* < 0.0001; Figure 4). However, when the two least stable reference genes (*RPL13* and *RPS16*) were selected, no significant difference was observed in the relative expression of *HSP70* between the head and legs (F_3,16_ = 9.1, *p* = 0.001; Figure 4). For another gene, the relative expression trend of *GR21* in different tissues was inconsistent in the two reference-gene combinations (*RPL8* and *RPL11, RPL13* and *RPS16*). The relative expression of *GR21* in the head was the highest, and the relative expression in the four tissues was significantly different, when normalized by *RPL8* and *RPL11* (F_3,8_ = 439.9, *p* < 0.0001; Figure 5). While, when normalized by *RPL13* and *RPS16*, the tissue with the highest relative expression of *GR21* was the thorax. However, there was no significant difference in the relative expression among the head, thorax, and legs; in addition, no significant difference was observed among the head, abdomen, and legs (Figure 5). Therefore, these results further indicate that the expression levels of the target genes will be different when normalized by different reference-gene combinations.

## 4. Discussion

In the present study, the expression-level stability of 10 candidate reference genes in *C. hemipterus* under various experimental conditions was evaluated through qRT-PCR. According to the final evaluation results of RefFinder, it was found that the stable expressed reference genes recommended for different developmental stages, different tissues of the adult males, different sexes, different temperature treatments, and bed bugs stimulated by different gases varied. Among them, *RPL8* had the least variable expression levels under the five experimental conditions, which was consistent with the report of Zhu et al. [34]. In the research of Zhu et al. [34], the stability of four reference genes (*RPL11*, *RPL8*, *RPS16*, and *HSP70*) under the conditions of different populations, different developmental stages, different tissues, and dsRNA injected in *C*. *lectularius* was compared; that study also identified *RPL8* as an optimal reference gene. We infer that this consistency may be related to the close genetic relationship between *C*. *hemipterus* and *C. lectularius*. Another report about the selection of reference genes in *C. lectularius* showed that *RPL18* was defined as the best reference gene for various tissues and developmental stages of pesticide-exposed and susceptible strains, and their candidate reference genes include *Actin*, *β*-*tubulin*, *EF1α*, *GAPDH*, *syntaxin* (*SYN*), *RPL18*, and *ubiquitin*-C (*UBC*) [33]. However, our study did not include *RPL18* as a candidate reference gene. Whether there was a difference in the stability of *RPL8* and *RPL18* in *C. hemipterus* requires further study.

*Actin* and *GAPDH*, the most widely used internal reference genes in animals, insects, and plants, did not show good expression stability in *C. hemipterus* (Figure 2, Table 3). This significant variance in the expression stability has been reported in many previous studies [6,8,9,33,40,41]. *EF1α* [33,42] and *NADH* [43] also have been widely used as reference genes in different species and experimental conditions. In our experiments, *EF1α* was one of the top two most stable genes in the developmental-stage sample set, but it did not perform well among the other four factors. Similarly, *NADH* was one of the top two most stable genes in different temperature treatment samples; however, its stabilization was greatly reduced under the conditions of different developmental stages and sexes. These results further demonstrate that no single reference gene is suitable for all species and different treatment conditions.

The ribosomal protein (RP) is involved in translation and protein synthesis and is one of the most highly conserved proteins in all biological cells. The RP coding gene was considered to be one stable expressed reference gene and had been widely used in insect molecular research to regulate the level of gene expression over the past 10 years [44,45,46,47,48]. These genes include *RPL8* under the conditions of different populations, different developmental stages, different tissues, dsRNA injected in *C. lectularius* [34], different geographic populations, across developmental stages, different wing dimorphisms, different temperature treatments, and different antibiotic treatments of *Metopolophium dirhodum* (Walker) (Hemiptera: Aphididae) [43]; *RPL13* in the different tissues of *Rhopalosiphum padi* L. (Hemiptera: Aphididae) [49]; and *RPL11* in the conditions of different developmental stages and temperatures in *Aphis craccivora* Koch (Hemiptera: Aphidiae) [50]. In this study, the *RPL8* gene exhibited the most stable expression level in *C. hemip**terus* under the five experimental conditions, *RPL11* was recommended as one of the top two most stable expressed genes in different adult tissues, and *RPS16* was also recommended for different sexes. Our results further expand the application scope of the RP coding gene as an internal reference gene in hemiptera insects.

It has been widely accepted that introducing two or more reference genes to normalize target-gene expression levels can reduce various errors and ensure the accuracy of results in experiments [51,52]. However, using a poor stable reference gene may yield the incorrect expression pattern of the target gene and lead to erroneous interpretations [50]. In the present experiment, the expression level of *GR21* was inconsistent among different tissues when normalized to the top two most stable (*RPL8* and *RPL11*) and least stable (*RPL13* and *RPS16*) reference genes. Therefore, it is pivotal to choose and confirm the best stable reference genes for certain insect species under specific conditions.

Over the past 20–30 yr period, there was a significant upward trend of bed bug infestations worldwide. The effective control of *C. hemipterus* has become a common issue in many infested countries or regions, such as China [19,20,53], southeast Asia [54,55], and South Africa [56]. Our results can provide the essential information for target-gene expression analysis and functional studies of this pest in order to lay a foundation for further exploration of new *C. hemipterus* control methods/strategies. However, this study only screened the reference genes of *C. hemipterus* under three biotic (developmental stage, tissue, and sex) and two abiotic (gas stimulation and temperature) conditions. Whether the results are also suitable for other experimental conditions (such as RNAi, insecticide treatment, etc.) in *C. hemipterus* or other species needs to be further verified.

## 5. Conclusions

In this study, we selected 10 frequently used reference genes in insects and evaluated their stability in *C. hemipterus* under various biotic (developmental stage, sex, and tissue) and abiotic (gas stimulation and temperature) conditions using RefFinder. The results show that the most suitable candidate combinations of reference genes were as follows: *RPL8* and *EF1α* for the developmental stage (eggs, early instar nymphs, late instar nymphs, and adults), *RPL8* and *RPS16* for adult sex, *RPL8* and *RPL11* for adult tissue (head, thorax, abdomen, and legs), *RPL8* and *β-tubulin* for gas stimulation (air and carbon dioxide), and *RPL8* and *NADH* for temperature treatment. These findings will provide the basic information for subsequent target-gene-expression analysis and functional studies in *C. hemipterus*.

## Figures and Tables

**Figure 1 insects-13-00784-f001:**
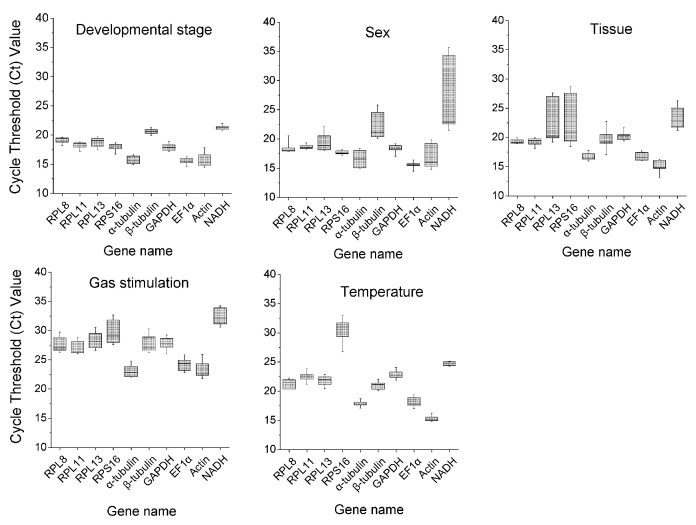
Cycle threshold (Ct) values of 10 candidate reference genes in *C. hemipterus* under different experimental conditions. The boxes show the 25th and 75th percentiles, the line inside the box depicts the median value, and whiskers indicate the standard error of the mean.

**Figure 2 insects-13-00784-f002:**
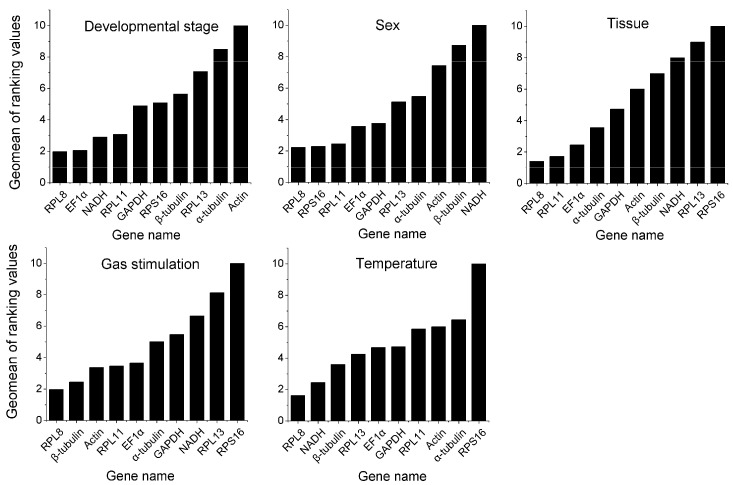
Stability ranking of 10 candidate reference genes in *C. hemipterus* under different treatment conditions analyzed by RefFinder (integrates four major computational programs: geNorm, NormFinder, BestKeeper, and ∆Ct). The lower the value, the higher the stability.

**Figure 3 insects-13-00784-f003:**
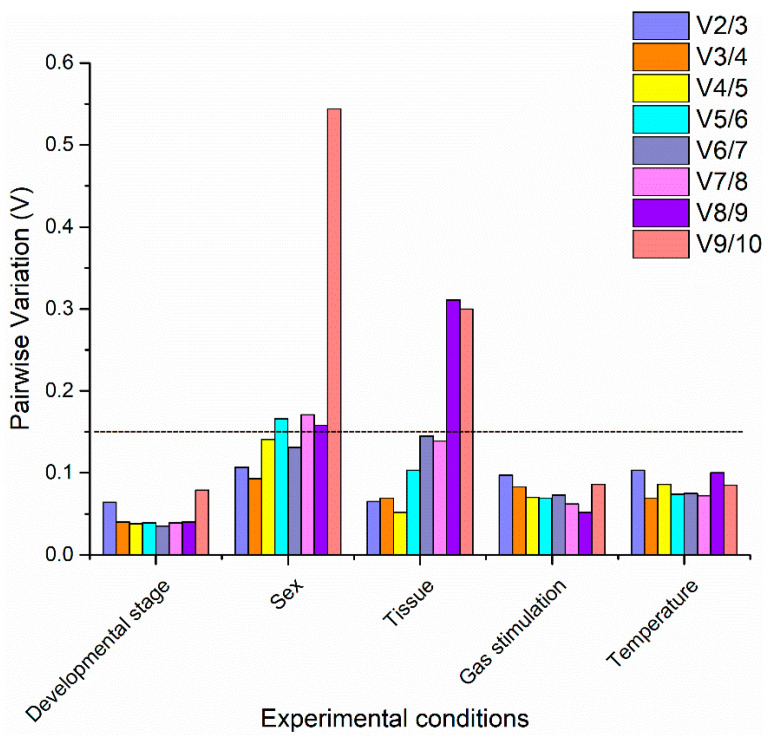
Optimal reference-gene number was required for the normalization of target-gene expression in *C. hemipterus* under the different experimental conditions. The pairwise variation (V) value was calculated using geNorm to evaluate the optimal reference-gene number in the qRT-PCR analysis. A value less than 0.15 indicates that n reference genes are sufficient to normalize the expression of the target gene.

**Figure 4 insects-13-00784-f004:**
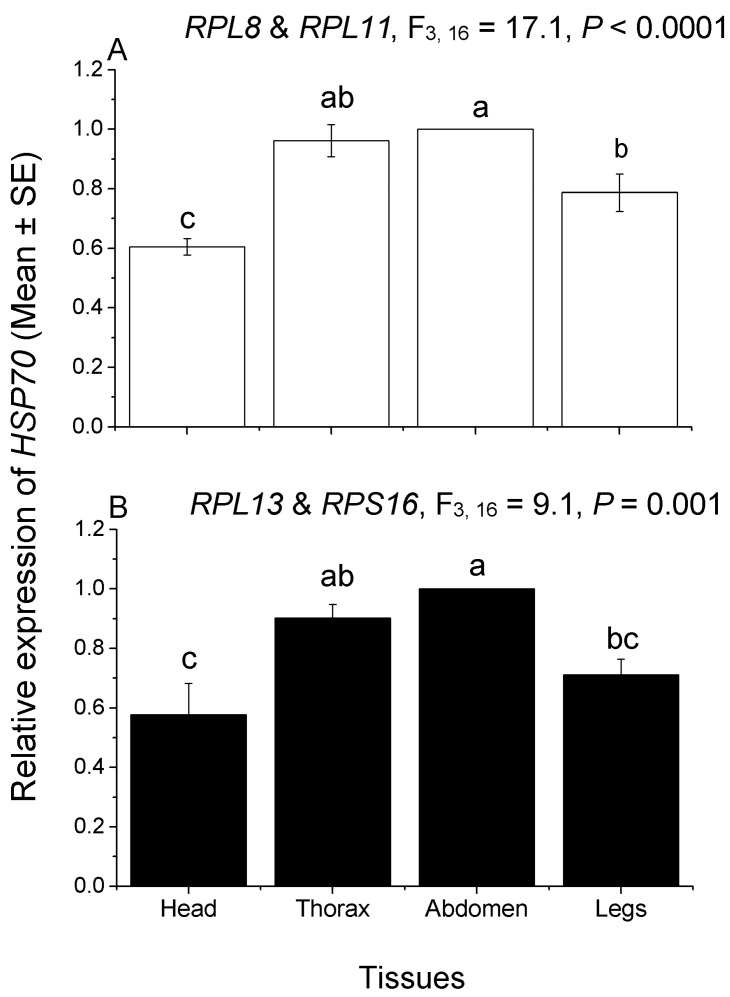
Tissue expression patterns of *HSP70* gene in *C. hemipterus*. The relative expression levels of *HSP70* in the head (including antennae and mouthparts), thorax, abdomen, and legs were normalized to the top two most stable (**A**, *RPL8* and *RPL11*) and least stable (**B**, *RPL13* and *RPS16*) candidate reference genes, respectively. Different lower letters above bars represent significant differences in gene expression among different tissues (*p* < 0.05, Tukey’s HSD test).

**Figure 5 insects-13-00784-f005:**
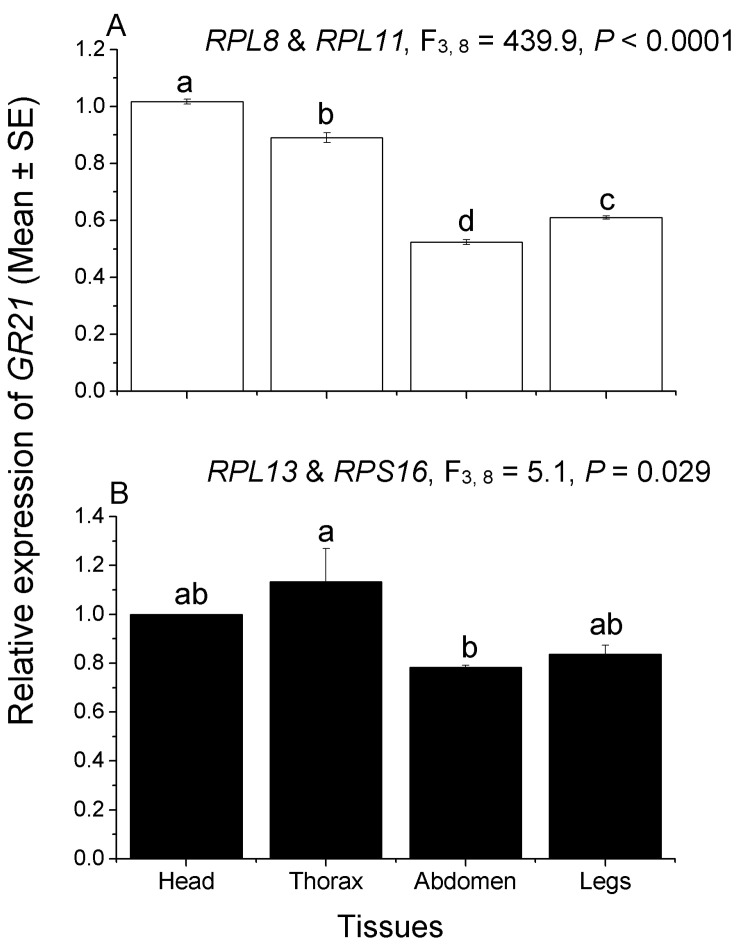
Tissue expression patterns of *GR21* gene in *C. hemipterus*. The relative expression levels of *GR21* in the head (including antennae and mouthparts), thorax, abdomen, and legs were normalized to the top two most stable (**A**, *RPL8* and *RPL11*) and least stable (**B**, *RPL13* and *RPS16*) candidate reference genes, respectively. Different lower letters above bars represent significant differences in gene expression among different tissues (*p* < 0.05, Tukey’s HSD test).

**Table 1 insects-13-00784-t001:** The primer sets and amplification efficiency of the 10 candidate reference genes used for qRT-PCR.

Gene	Primer Sequence ^a^ (5′–3′)	Amplicon Length (bp)	Efficiency ^b^ (%)	R^2 c^
*RPL8*	F: AGGCACGGTTACATCAAAGG	131	111.07	0.9952
	R: TCGGGAGCAATGAAGAGTTC			
*RPL11*	F: GAAGAATGTCATGCGAGATGTCAGG	129	92.28	0.9987
	R: CCTTTGAGAAGACTGGCTGCTG			
*RPL13*	F: ATTGGCAGAGGTTCATTCGT	131	100.91	0.9979
	R: GCATCTCACTGCTGGTCTCA			
*RPS16*	F: AATGGTCGCCCACTTGAGAT	158	87.46	0.9976
	R: GCCTGCCTGATTGCGTAGAT			
*α-tubulin*	F: TCAACTACCAACCACCCACT	139	94.93	0.9990
	R: TTGGCGTACATCAAGTCAAA			
*β-tubulin*	F: CCTTTGCTGGACGCCTCATT	84	103.67	0.9902
	R: CGACCCGACTGGTGCATACC			
*GAPDH*	F: GCATTTAGAGTCCCTGTCGC	149	91.12	0.9806
	R: ACTTCATCTTCGGTGTAGCC			
*EF1α*	F: CGTTAGGACGTTTCGCTGTC	103	106.65	0.9893
	R: GCCTTTGTCACTTTGCCACT			
*Actin*	F: GACTTCGAGCAGGAAATGGC	114	98.08	0.9989
	R: TTCTGGGCAACGGAACCTCT			
*NADH*	F: AAAGACGAAGGTCGGGATTA	76	87.11	0.9812
	R: GCATCAAAGAAACACGCATC			

^a^: F and R denote forward and reverse primers, respectively. ^b^: qRT-PCR amplification efficiency, calculated by the standard curve method. ^c^: Determination coefficient.

**Table 2 insects-13-00784-t002:** Cycle threshold (Ct) values (mean ± SE) of 10 candidate reference genes under five different experimental conditions.

Candidate Gene	Experimental Conditions					All *
	Developmental Stage	Sex	Tissue	Gas Stimulation	Temperature	
*RPL8*	19.15 ± 0.16	18.50 ± 0.42	19.33 ± 0.11	27.61 ± 0.56	21.26 ± 0.23	20.68 ± 0.40
*RPL11*	18.23 ± 0.57	18.62 ± 0.16	19.29 ± 0.16	26.97 ± 0.49	22.39 ± 0.23	20.78 ± 0.41
*RPL13*	18.75 ± 0.24	19.40 ± 0.66	22.24 ± 1.03	28.34 ± 0.60	21.53 ± 0.23	21.60 ± 0.48
*RPS16*	17.94 ± 0.65	17.63 ± 0.15	22.73 ± 1.19	29.70 ± 0.87	30.44 ± 0.41	24.09 ± 0.83
*α-tubulin*	15.74 ± 0.67	16.59 ± 0.61	16.58 ± 0.16	23.08 ± 0.47	17.67 ± 0.16	17.47 ± 0.33
*β-tubulin*	20.61 ± 0.44	22.18 ± 0.97	19.50 ± 0.50	27.67 ± 0.65	20.95 ± 0.17	21.46 ± 0.38
*GAPDH*	17.88 ± 0.53	18.30 ± 0.32	20.25 ± 0.20	27.77 ± 0.48	22.67 ± 0.19	21.06 ± 0.45
*EF1α*	15.54 ± 0.54	15.52 ± 0.25	16.70 ± 0.17	24.27 ± 0.46	17.94 ± 0.23	17.55 ± 0.39
*Actin*	15.67 ± 1.20	16.87 ± 0.88	14.92 ± 0.31	23.29 ± 0.63	15.42 ± 0.12	16.46 ± 0.40
*NADH*	21.26 ± 0.35	26.60 ± 2.67	23.12 ± 0.52	32.34 ± 0.60	24.61 ± 0.10	24.61 ± 0.56

*: The mean cycle threshold (Ct) values (mean ± SE) of each reference gene under five experimental conditions in *C. hemipterus*.

**Table 3 insects-13-00784-t003:** Stability of 10 reference genes in *Cimex hemipterus* under different conditions.

Condition	Rank *	geNorm		NormFinder		BestKeeper		∆Ct	
		Gene	Stability	Gene	Stability	Gene	Stability	Gene	Stability
Developmental stage	1	*RPL8*	0.118	*EF1α*	0.274	*NADH*	0.283	*RPL8*	0.622
	2	*RPL11*	0.118	*NADH*	0.278	*β-tubulin*	0.349	*EF1α*	0.623
	3	*EF1α*	0.240	*RPL8*	0.334	*EF1α*	0.410	*RPL11*	0.636
	4	*RPS16*	0.269	*GAPDH*	0.378	*GAPDH*	0.428	*RPS16*	0.675
	5	*RPL13*	0.356	*RPL11*	0.382	*RPL8*	0.443	*NADH*	0.707
	6	*GAPDH*	0.434	*RPS16*	0.433	*RPL11*	0.457	*GAPDH*	0.722
	7	*NADH*	0.497	*β-tubulin*	0.588	*RPS16*	0.490	*RPL13*	0.837
	8	*α-tubulin*	0.560	*RPL13*	0.685	*α-tubulin*	0.622	*β-tubulin*	0.865
	9	*β-tubulin*	0.629	*α-tubulin*	0.708	*RPL13*	0.672	*α-tubulin*	0.896
	10	*Actin*	0.813	*Actin*	1.513	*Actin*	0.963	*Actin*	1.551
Sex	1	*RPS16*	0.328	*RPL8*	0.364	*RPS16*	0.290	*RPL8*	1.709
	2	*EF1α*	0.328	*RPL13*	0.428	*RPL11*	0.317	*RPL11*	1.794
	3	*RPL11*	0.640	*RPL11*	0.709	*EF1α*	0.385	*GAPDH*	1.914
	4	*GAPDH*	0.749	*GAPDH*	1.133	*GAPDH*	0.520	*RPS16*	1.944
	5	*RPL8*	0.881	*α-tubulin*	1.192	*RPL8*	0.691	*α-tubulin*	1.947
	6	*α-tubulin*	1.002	*Actin*	1.217	*α-tubulin*	1.253	*EF1α*	1.948
	7	*RPL13*	1.183	*RPS16*	1.380	*RPL13*	1.272	*RPL13*	2.039
	8	*Actin*	1.325	*β-tubulin*	1.428	*Actin*	1.771	*Actin*	2.091
	9	*β-tubulin*	1.421	*EF1α*	1.448	*β-tubulin*	1.986	*β-tubulin*	2.241
	10	*NADH*	2.388	*NADH*	6.188	*NADH*	5.589	*NADH*	6.256
Tissue	1	*RPL8*	0.355	*RPL8*	0.178	*RPL8*	0.329	*EF1α*	1.595
	2	*RPL11*	0.355	*RPL11*	0.178	*α-tubulin*	0.415	*RPL8*	1.600
	3	*EF1α*	0.484	*EF1α*	0.247	*RPL11*	0.422	*RPL11*	1.667
	4	*GAPDH*	0.556	*α-tubulin*	0.312	*EF1α*	0.490	*α-tubulin*	1.723
	5	*α-tubulin*	0.632	*GAPDH*	0.579	*GAPDH*	0.524	*GAPDH*	1.775
	6	*Actin*	0.847	*Actin*	0.830	*Actin*	0.808	*Actin*	1.915
	7	*β-tubulin*	1.057	*β-tubulin*	1.165	*β-tubulin*	1.173	*β-tubulin*	2.158
	8	*NADH*	1.313	*NADH*	2.099	*NADH*	1.524	*NADH*	2.656
	9	*RPL13*	1.838	*RPL13*	3.835	*RPL13*	3.203	*RPL13*	4.031
	10	*RPS16*	2.353	*RPS16*	4.235	*RPS16*	3.629	*RPS16*	4.412
Gas stimulation	1	*Actin*	0.325	*RPL8*	0.241	*EF1α*	0.893	*RPL8*	0.689
	2	*β-tubulin*	0.325	*β-tubulin*	0.400	*GAPDH*	0.958	*β-tubulin*	0.737
	3	*RPL8*	0.440	*RPL11*	0.435	*α-tubulin*	0.998	*RPL11*	0.764
	4	*RPL11*	0.491	*Actin*	0.453	*RPL11*	1.004	*Actin*	0.766
	5	*α-tubulin*	0.588	*NADH*	0.579	*RPL8*	1.100	*EF1α*	0.834
	6	*EF1α*	0.624	*EF1α*	0.616	*RPL13*	1.105	*α-tubulin*	0.846
	7	*GAPDH*	0.644	*α-tubulin*	0.651	*NADH*	1.180	*NADH*	0.857
	8	*NADH*	0.696	*GAPDH*	0.718	*Actin*	1.231	*GAPDH*	0.892
	9	*RPL13*	0.778	*RPL13*	0.855	*β-tubulin*	1.281	*RPL13*	1.041
	10	*RPS16*	0.864	*RPS16*	1.089	*RPS16*	1.767	*RPS16*	1.207
Temperature	1	*RPL8*	0.230	*RPL8*	0.279	*NADH*	0.345	*RPL8*	0.616
	2	*RPL13*	0.230	*β-tubulin*	0.343	*Actin*	0.390	*NADH*	0.653
	3	*EF1α*	0.360	*NADH*	0.361	*α-tubulin*	0.446	*β-tubulin*	0.660
	4	*RPL11*	0.416	*GAPDH*	0.407	*β-tubulin*	0.554	*EF1α*	0.664
	5	*GAPDH*	0.464	*EF1α*	0.426	*GAPDH*	0.563	*GAPDH*	0.679
	6	*NADH*	0.512	*RPL13*	0.447	*RPL11*	0.726	*RPL13*	0.688
	7	*β-tubulin*	0.528	*RPL11*	0.462	*RPL8*	0.764	*RPL11*	0.702
	8	*Actin*	0.573	*α-tubulin*	0.568	*EF1α*	0.776	*α-tubulin*	0.791
	9	*α-tubulin*	0.605	*Actin*	0.593	*RPL13*	0.797	*Actin*	0.795
	10	*RPS16*	0.766	*RPS16*	1.341	*RPS16*	1.211	*RPS16*	1.411

*: The ranking is based on stability; the lower the value, the higher the ranking.

## Data Availability

The data presented in this study are available on request from the corresponding author.

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
