# Peer review of "Identification and Validation of Reference Genes for Expression Analysis Using qRT-PCR in Cimex hemipterus (Hemiptera: Cimicidae)"

_insects, 2022, doi:10.3390/insects13090784_

Round 1

Reviewer 1 Report

In this manuscript, the authors intended to validate ten common reference genes. I think that it is a beneficial attempt, however, their conclusion seems to be somewhat misleading. Strictly speaking, the results described in the manuscript is only applicable to the primer-pairs used in this experiment. The stability or instability might be due to not the expression-level of the genes but the effectiveness of sequences of the primer pairs. Thus, the authors should show a clear evidence to exclude the possibility that the adequacy of the primer-design does not  affect this validation.

Author Response

Response: Many thanks for your suggestions and comments. Your suggestion is valid and reasonable. In fact, the primers used in our experiments were designed strictly according to the principles of qRT-PCR primer design using Primer Premier 5 software. When selecting primer-pairs, we preferentially select the primers with the highest score, and do not to select primers whose primers themselves may form dimers or hairpin structures. In the preliminary experiment, we have evaluated the feasibility of primers carefully, and results indicated that the primers used in this experiment were specific and reliable. Therefore, they are fully suitable for reference gene selection in tropical bed bugs.

Reviewer 2 Report

It is a pleasure to review the work “Identification and Validation of Reference Genes for Expression Analysis Using qRT-PCR in Cimex hemipterus (Hemiptera: Cimicidae)” that included basic science which is absolutely necessarily to develop effective strategies to control the bed bug pest, which is a non-model organism. In this work, the authors inform us which genes researchers can use as reference genes in order to do gene expression analysis by qRT-PCR specifically in C. hemipterus. They found two candidate genes as best reference genes base on expression-level stability in different conditions. They suggested that researchers use these two reference genes to give better results in the normalization of the target gene and they give an example using HSP70 as target gene. They mentioned that using different combinations of reference genes will give different expression levels of the target gene which is one of the challenges that a researcher needs to deal with. Here the authors give information about which reference genes can be used under biotic and abiotic conditions. However, I would like to see more biological replicates in the target HSP70 gene’s experiment because the results of expression levels of HSP70 between different tissues using the top two most stable and least stable candidates references genes looks pretty similar. I suggest using at least one more target gene to give more support to the conclusion also. On the other hand, it might be very interesting to use ssRNA and dsRNA GFP as negative control to validate the references gene since one of the strategies to deal with the pests focuses on the use of RNAi mechanism and given that the precursors of the small RNA can trigger physiological responses that could affect the expression levels of the references genes.

Author Response

Response: Many thanks for your suggestions and comments, they are very helpful to improve our manuscript. In the revised manuscript, two additional biological replicates were added in the target HSP70 gene’s experiment; in addition, the validation of another target gene (GR21) in different tissues was also determined. However, for the ssRNA and dsRNA GFP treatment, we did not add in the revised manuscript. Because we have not synthesized double-stranded RNA and ssRNA at present. Maybe it could be conducted in the further study, we have mentioned this in the revised manuscript. Nonetheless, it is worth noting that the five experimental conditions used in this study were selected under random conditions, which are sufficient for the screening of reference genes in tropical bed bugs.

Round 2

Reviewer 1 Report

In the previous review, I misunderstood that the authors were going to evaluate the utility of the ten reference genes in qRT-PCR universally applicable  across species. Now, it is clear that their finding is limited to functional studies in C. hemipterus. In my opinion, it is unlikely that the manuscript is informative to the readers who do not study C. hemipterus.    

Author Response

Response to Reviewer 1 Comments

Point 1: In the previous review, I misunderstood that the authors were going to evaluate the utility of the ten reference genes in qRT-PCR universally applicable across species. Now, it is clear that their finding is limited to functional studies in C. hemipterus. In my opinion, it is unlikely that the manuscript is informative to the readers who do not study C. hemipterus.

Response: Many thanks for your suggestions and comments. Reference genes are the basis for exploring the function of target genes in insects. So far, there is no report on the screening and application of the reference genes in C. hemipterus. The objective of our study was to screen the stably expressed reference genes of C. hemipterus under five experimental conditions for qRT-PCR normalization so that we can further study the expression patterns of target genes and clarify their functions in C. hemipterus. In addition, we also hope that this research can help to better understanding this pest and formulate reasonable control strategies/methods. In the last paragraph of discussion, we add some information about the global bed bug infestation and reiterate the significance of the reference gene screening in this study.

In addition, some other modifications were also conducted in the revised manuscript, and we have showed them with track changes. Please kindly find them.